# Do Dynamic Capabilities and Digital Transformation Improve Business Resilience during the COVID-19 Pandemic? Insights from Beekeeping MSMEs in Indonesia

Jaisy Aghniarahim Putritamara [1,*], Budi Hartono [1], Hery Toiba [2], Hamidah Nayati Utami [3], Moh Shadiqur Rahman [2] and Dewi Masyithoh [4]

1   Department of Livestock Socio-Economics, Faculty of Animal Science, University of Brawijaya, Malang 65145, Indonesia
2   Socioeconomics Department, Faculty of Agriculture, University of Brawijaya, Malang 65145, Indonesia
3   Faculty of Business Administration, University of Brawijaya, Malang 65145, Indonesia
4   Department of Livestock Socio-Economics, Faculty of Animal Science, Islamic University of Malang, Malang 65144, Indonesia
*   Correspondence: jaisyap@ub.ac.id

**Abstract:** This study investigated the influence of Dynamic Capabilities (DC) and Digital Transformation (DT) on Business Resilience During the COVID-19 Pandemic. Using cross-sectional data from 388 micro, small, and medium enterprises (MSMEs) of beekeeping in Indonesia. Furthermore, the data were analyzed by Structural Equation Model (SEM) analysis and executed by SmartPLS 3.0 software. The main results of this study indicate that DC plays an important role in improving MSMEs' DT. However, the essential role of DT on firm resilience only happened for micro, small, and medium firms of family businesses. However, DT has an insignificant effect on firm resilience in small nonfamily businesses. Yet, the effect of DC on firm resilience is mediated by DT. Nevertheless, our empirical findings indicate heterogeneous effects among micro, small, and medium firms. Based on the study's findings, we suggested that the policy implication in developing beekeeping firms should be more specific based on the firm scale. The results of this study can be generalized to the national level to inform decision-making regarding the intangible assets of MSME livestock products in developing countries. The findings are also relevant to other livestock products, which tend to be dynamic during a crisis.

**Keywords:** dynamic capabilities; digital transformation; business resilience; beekeeping; Indonesia

## 1. Introduction

The economic turmoil caused by the COVID-19 pandemic in the past three years has threatened the sustainability of businesses of all sizes in both developed and developing countries [1–5]. For example, China saw a 13–24% decrease in the investment value of micro, small, and medium enterprises (MSMEs) and a 6.2% increase in unemployment [5,6]. Likewise, Romania saw a decrease in the value of the net income of its large and small companies by 37% [7]. The economic slowdown was caused by the disruption in the supply chain, whose operations were obstructed by the lockdown policies. By Harris, et al. and Moosavi, et al. [8,9] sales plunged by 80%, and 60% of people had difficulty accessing food products. The most vulnerable sector that was affected by COVID-19 is livestock, such as beekeeping. Whereas beekeeping helps increase food security and contributes positively to revenue in Indonesia, the government and society are still paying no notice. Indonesia is considered to be a net importer of honey (mostly from Asia). More specifically, honey exports in 2013 totaled 207 tons and 2.35 million USD, while imports of 2.177 tons of honey totaled 8.33 million USD [10]. The price of premium honey sold on the domestic market in 2018 was 200,000 Rupiahs (about 14 USD), according to UNPAD. In Indonesia, expanding

the beekeeping industry improves environmental protection while also raising the standard of living and revenue. Marketing bees could raise people's living levels.

An adequate supply chain system is a predictor of business continuity [11], and supply chain problems pose a threat to businesses, with the worst case being bankruptcy [12]. Therefore, businesses race to increase resilience. [13–15]. Basically, resilience is a dynamic growth process that is linked to the characteristic of personality [16]. Resilience, as defined by Walker et al. [17], is the capacity to deal with disruption or the capacity to hold on to the components required for updating or rearranging a system's functionality. A business resiliency would constantly look for opportunities to take risks and profit from circumstances. Yet, resilience is also linked to foreseeing and averting unanticipated risks [18]. Additionally, it's critical to be sensitive, manage a fluid decision-making process, and alter views. Organizational agility and resilience are needed in times of economic instability and commercial disruption, and resilience is best shown after an incident or catastrophe. Businesses that have high levels of business resilience can promptly respond to disturbances while protecting their people, assets, and overall brand equity [19]. Fiksel [20] pointed out that business resilience is the "ability of organizations to survive, adapt, and develop in the face of chaotic change". Dahles et al. [21] suggested that the ability of businesses to adapt to sudden changes and shocks is essential for economic growth. Businesses that are resilient are able to bounce back from setbacks and exhibit adaptability, which can result in significant modifications to the overall business model [20]. Smaller companies are more flexible and adaptive than larger ones, making them more sensitive to exogenous shocks.

Researchers in this field have focused more on resilience in dealing with shocks, such as the impact of the COVID-19 pandemic. For example, Mangalaraj, et al. [22] show how the adverse effects of the pandemic have forced retail businesses to transform their technology and information (IT) capabilities, often referred to as digital transformation (DT), to be more adaptive to turbulence. Likewise, Elgazzar, et al. [23] argue that DT is the key to increasing business resilience and long-term profitability. Companies need to be flexible in dealing with rapid changes, including consumer behavior. In this case, DT can help respond to external stimuli fast. Also, DT stimulates and allows for innovation that make business capable of changing their business model to add value to customers. In other words, DT can allow companies to achieve dynamic capabilities.

Past research has shown a direct relationship between DC and DT. For example, Warner et al. [24] explain that DC is a prerequisite for businesses to build resilience. Meanwhile, DT promotes collaboration through co-creation and competition [24]. Together, DC and DT allow business people to create new business models and be more agile in adapting to changes with the resources they have. Aside from strengthening DC and DT, balancing external and internal collaboration is necessary to create a flexible and conducive organizational culture. The role of DC in encouraging the maturity of DT is supported by a study by Soluk et al. [25] on family-owned manufacturing companies in Germany, Austria, and Switzerland.

Previous studies have also proved that DC supports resilience. For example, a study by Ozanne et al. [26] on 419 MSMEs in Australia and New Zealand showed that MSMEs need to invest in DC to bounce back from the COVID-19 pandemic coupled with other resources capable of increasing resilience. Meanwhile, in the tourism sector, Jiang, et al. [27] explain that to achieve organizational resilience, (1) changes in operational routines are needed, (2) DC is necessary to compensate for the lack of resources for these changes to happen, (3) DC can also balance out the limited internal and external to deal with environmental threats, (4) internal and external resources need to be integrated, (5) operational routines need to be adjusted with market situations, and (6) sector, size, and age are predictors of resilience.

Previous research has also shown that DT is needed as a mediator to achieve resilience. For example, Zhou et al. [28] studied 3213 companies in China and found that DT is a mediator that bridges the executives' confidence and environmental technology innovation, especially in the face of fierce competition and economic uncertainty. The study also shows

that confident executives are more optimistic about creating an innovative environment, so they initiate DT and are ready to face the challenges of establishing DT.

In sum, the direct impact of DC and DT has been documented in past studies in international contexts, mainly using a qualitative approach. Quantitatively, the impact has been observed by Songkajorn, et al. [29] in the auto-parts industry. Specific to DT, the impact on resilience has been proven in the retail industry in developed countries [30]. The indirect impact has also been observed using a mixed-method approach [31]. As for DC, a direct influence on resilience has been observed in MSMEs in the tourism sector in developed countries [26,27].

In terms of indirect impact, a previous study by Zhou et al. [28] did not look into more details of the role of DC in mediating DT and resilience, although they observed the mediator role of DT. In other words, no studies have proven the role of DT in mediating DC and resilience. Previous studies have also not discussed MSME livestock products, such as honeybees, as these products tend to be dynamic in the face of climate change and market changes during a shock like the COVID-19 pandemic. The demand increased, but the supply was low [32] as the movement restrictions disrupted distribution activities [33].

Therefore, based on the existing literature, there are three essential research gaps that are open for exploration. First, there is no evidence concerning the association among DC, DT, and firm resilience of beekeeping, especially in developing countries such as Indonesia. Second, the existing literature overlooked the firm size categories (micro, small, or medium business), and business institutional (Family or non-family business). Third, this study makes the first attempt to explain the mediation effect of DT.

Therefore, to fill these gaps, this research aims to examine the impact of DC on DT, DT on resilience, DC on resilience, and the role of DT in mediating the impact of DC on resilience based on firm size and institutional category. This research contributes to the literature two-fold. First, this study provides the first empirical evidence of the influence of DC on resilience mediated by DT in the case of beekeeping MSMEs during the COVID-19 pandemic. Second, we provide a disaggregated estimation in our model based on firm size and business types, whether a family business (FB) or a non-family business (NFB). The findings of this research are to have implications for honey product SMEs in dealing with the COVID-19 pandemic shock.

## 2. Theoretical Background and Hypothesis Development

### 2.1. The Link between Dynamic Capabilities (DC) and Digital Transformation (DT)

A review by Vial [34] examining 282 past studies defines DT as continuous changes and disruptions to business that make the environment hypercompetitive and force businesses to adapt. In this case, DC allows companies to adapt to waves of technological innovation through environmental scanning, sensing, and integrative capabilities. DC enables companies to achieve business performance geared towards strategic change.

Another study using a qualitative approach by Warner et al. [24] collected data from interviews with senior company consultants in 2017–2018. The results show that the role of DT includes (1) reconstructing the old business model, (2) strengthening the collaborative approach, and (3) building a collaborative culture. The variables used to measure the mediating role of DC in triggering, encouraging, and inhibiting DT include (1) digital sensing, including scouting, scenario planning, and mindset crafting, (2) digital seizing, including rapid prototyping, digital portfolio balancing, and agility, (3) navigating innovation ecosystem, redesigning internal structures, and improving digital maturity. However, the study only involved companies in developed countries, so it may not be relevant to those in developing countries.

Another study by Magistretti et al. [35] initiated design thinking as part of DC, which then encouraged DT to (1) extend collaboration with external parties to increase knowledge, (2) synchronize technology with HR perspectives to adopt technological changes to HR capacity, (3) crop and utilize technology according to the features needed, (4) interpret the ability to identify new opportunities and (5) recombine technology with all resources.

Another study by Songkajorn et al. [29] examined the auto parts industry in Thailand using a quantitative approach and showed that DC had a positive effect on DT. This study introduces knowledge-based DC, which includes absorption, generation, storage, and adaptation. Absorption and generation capability encourage entrepreneurs to continuously acquire new knowledge, while storage capability allows them to gain knowledge quickly. However, this research uses a qualitative approach [24,25,35,36]. Therefore, this study offers a novelty by connecting DC and DT in the case of food and livestock MSMEs in developing countries.

**H1:** *DC positively impacts DT.*

### 2.2. The Link between Digital Transformation (DT) and Business Resilience

A study by He et al. [37] involving 474 MSMEs in the service sector shows that strategic technology allows companies to continue business operations. DT helps employees navigate business turbulences by actively seeking resources and quickly developing adaptive solutions. DT also helps companies achieve vision, governance, culture, and leadership that can encourage employees to continue to innovate in the face of a crisis.

In addition, an empirical study by Zhang et al. [31] observed 339 companies in China and shows that DT has a positive effect on resilience. DT plays a vital role in innovatively driving resilience. DT can encourage companies to continue to explore internal business factors and prompt innovation, making it resilient in the long term. At its core, DT is a company's innovation process to deal with uncertainty. Leaders with DT must have the ability to think digitally, have knowledge of digitalization, and appreciate the internal and external environment.

A literature review by Elgazzar et al. [23] shows that DT positively impacts resilience. Therefore, companies applying DT can increase long-term business profitability. Khurana, et al. [38] studied MSMEs with a qualitative case-study approach in India's manufacturing and service sectors during the pandemic. This study measures a resilience model at three business sizes: micro, meso, and macro. The results show that DT is needed to achieve resilience as it improves resource management. In times of a crisis like the pandemic, DT can help businesses survive by navigating the changes responsively. Mangalaraj, et al. [22] show that, in retail companies, organizational dependence on IT encourages entrepreneurs to maintain corporate strategies as IT competence makes companies more agile and responsive to changes. As mentioned by Kazemi et al. [39] firms need to improve their ability to improve their competitiveness and their value to compete with their rivals, and it can be done by developing and improving their DT [23].

Based on the past empirical evidence and literature studies described above, it can be concluded that there is a relationship between DT and resilience. However, there has not been evidence on livestock products such as honeybees. The second aim of this study is to fill the gap by providing evidence from the beekeeping industry.

**H2:** *DT positively impacts resilience.*

### 2.3. The Link between Dynamic Capability (DC) Dan Business Resilience

Khurana et al. [38] used a qualitative approach with case studies of eight MSME entrepreneurs in India who had to change their business models during the pandemic. The study results show that three DC components are needed to achieve resilience: seizing, reconfiguring, and changing the model. Seizing allows entrepreneurs to identify the technology used by the market segments. Reconfiguring helps entrepreneurs to manage existing resources they own and outsource the rest. Also, as the absolute advantage and comparative advantage theory, firms need to maintain their profit and comparative advantage [40]. This can be achieved by having a better DC.

Kurtz et al. [41] argue that DC builds a competitive advantage through resource management and adaptive capabilities, encouraging MSMEs to achieve resilience. DC consists

of four components: (1) sensing, the ability to recognize opportunities, monitor the market, and analyze competitor changes to adjust resources to deal with instability, (2) learning, the capability to gain new knowledge to achieve perpetual competitive advantage and evaluate weaknesses and strengths, (3) integrating, the ability to apply knowledge to the internal business conditions to make strategic decisions, (4) coordinating, the ability to connect changes, knowledge, and resources. Entrepreneurs with sensing, seizing, and configuring skills can explore and exploit resources to build resilience. A study by Ozanne et al. [26] in Australia and New Zealand examined MSMEs using a quantitative approach with 199 respondents. The results show that DC positively affected resilience, as it helped them balance limited resources and unexpected business changes.

In the tourism sector, a quantitative study by Wided [42] in Saudi Arabia with 200 respondents shows that DC positively affects resilience. Likewise, DC in this context consists of (1) sensing the turbulence; (2) learning to gain new knowledge and using it as capital; (3) integrating the business with new opportunities in the form of new products, processes, and services; and (4) coordinating new resources with the existing resources.

In sum, research has shown how DC plays a crucial role in various contexts. However, the impact on resilience in food product MSMEs has not been explored. This study aims to fill this gap by examining the impact of DC on food products using a quantitative approach with multivariate analysis in the case of beekeeping MSMEs.

**H3:** *DC positively impacts resilience.*

### 2.4. The Role of DT as a Mediator Variable

An empirical quantitative study by Songkajorn et al. [29] focusing on MSME spare parts in Thailand shows that DT mediates the relationship between DC and Organizational Strategic Intuition (OSI). The value of the indirect effect is smaller than the direct effect of DT on OSI. This finding also illustrates how DT can be a strategy to be adopted to adapt to changing markets during turbulence. DT allows companies to introduce new products, processes, and services according to the changing customer needs while managing organizational structural changes.

Furthermore, an empirical quantitative study by Li et al. [43] examines the manufacturing industry in China. The results show that DT mediates the relationship between the digital economy and enterprise innovation. A similar result was found in the study by Zhou et al. [28] They conducted an empirical quantitative study involving companies in Shanghai and Shenzhen, China, using panel data from 2007 to 2019. The results show that DT mediates the relationship between executive high confidence levels and environmental innovation. This indicates that DT is driven by confidence. Challenges from DT adoption are worth tackling to face uncertain, hypercompetitive business environments. As such, entrepreneurs can continue to innovate.

Sousa-Zomer et al. [44] studied the role of DT in large companies across economic sectors. The results show that DT mediates (1) the relationship between digital intensity in the business processes and business performance and (2) between the conditions for action and interaction and performance. Organizations with solid DT capabilities have the foundation to deal with rapid changes, hence maintaining competitiveness. This is because digital business models can sustain and enhance business performance in an environment where digital intensity is needed, such as establishing digital partnerships with external parties.

In brief, research has shown how DC can function as a mediating variable. However, no studies have used a quantitative approach with multivariate analysis to prove the role of DC in mediating the link between DC and resilience. Meanwhile, DT has been proven to mediate variables that allow businesses to navigate uncertainty. Considering this background, this study examined beekeeping MSMEs in developing countries to fill the gap in the literature on the role of DT in mediating the link between DC and resilience.

**H4:** *DT mediating the link between DC and resilience.*

Figure 1 is the illustration of the hypothesis tests:

**Figure 1.** Research Framework.

### 3. Materials and Methods

*3.1. Measurement of Research Variables*

Three latent variables in this study are dynamic capability (DC), digital transformation (DT), and business resilience. The DC variable consists of sensing, seizing, and reconfiguring components, with a total of 29 indicators [26,45–50]. The DT variable consists of IT readiness and strategic alignment with 12 indicators [51–53]. The business resilience variable consists of robustness, readiness, response, and recovery with 21 indicators [26,54–56]. The questionnaire uses a Likert scale for all indicators, ranging from 1 for strongly disagree to 5 for strongly agree. The indicators for each latent variable are presented in Table 1.

**Table 1.** Research variables and indicators.

| | | Dynamic Capability (X) [26,45–50] |
|---|---|---|
| | 1 | I can identify new, more profitable technologies |
| | 2 | I have regular customers |
| | 3 | I can understand market conditions |
| | 4 | I always try to identify obstacles that make business processes inefficient |
| **Sensing** | 5 | I can identify new opportunities in the beekeeping business that competitors have not discovered |
| | 6 | I continue to develop beekeeping products to suit consumer needs |
| | 7 | I constantly look for ideas to improve the honey product quality |
| | 8 | I have to be proactive and reactive to business changes during the pandemic to anticipate challenges that threaten the sustainability |
| | 9 | I am willing to learn about new technology |
| | 10 | I continue to improve business strategies to take advantage of the current situation |
| | 11 | I am willing to spend money to solve pandemic-induced problems in the beekeeping business |
| | 12 | I maintained the best beekeeping business model during the pandemic |
| | 13 | I continue to increase my knowledge to develop beekeeping products |
| **Seizing** | 14 | I use knowledge to create new products |
| | 15 | I can formulate a business strategy |
| | 16 | I can secure strategic partnerships |
| | 17 | I can plan future investments in the beekeeping business |
| | 18 | I can analyze business feasibility (IRR/NPV, ROI, ROA, ROE) |
| | 19 | I can identify human resource requirements for the business |

**Table 1.** *Cont.*

| Reconfiguring | 20 | I can control and access beekeeping products' prices |
|---|---|---|
| | 21 | I can create new beekeeping products that are different from competitors |
| | 22 | I can respond and adapt to unexpected business changes |
| | 23 | I can invite business partners with high potential to work together, and I terminate partnerships with lower potential |
| | 24 | I can adapt the business processes to respond to changing business priorities |
| | 25 | I can change business processes to generate better profits |
| | 26 | I try to create effective and efficient communication in the beekeeping business |
| | 27 | I can grow employees' sense of responsibility to succeed in changing business plans for the better |
| | 28 | I can maintain consistency amid the changes in the beekeeping business caused by the pandemic |
| | 29 | I implement changes in the business plan to be flexible and adaptive to changes in the current business conditions |
| **Digital Transformation (Y)** [51–53] | | |
| IT Readiness | 1 | I use online platforms to store data related to the beekeeping business, such as Google Drive and Dropbox |
| | 2 | I use emails to support the beekeeping business |
| | 3 | I use software such as Microsoft Word, Excel, and PowerPoint to run a honey business |
| | 4 | I use social media (such as WA, Instagram, Facebook, TikTok, Twitter, and YouTube) to support my honey business |
| | 5 | I use the website to support the honey business |
| | 6 | I use the Google Analytics/Search Engine Optimization/Google Business/Social Media Analytics application to analyze honey sales |
| | 7 | I can integrate digital technology (WhatsApp, Instagram, Facebook, YouTube, data storage platforms (Google Drive), and data analytical tools (Google Analytics and social media analytics) in running a beekeeping business |
| Strategic Alignment | 8 | The role of digital technology, such as social media, storage platforms, and analytical tools, can change my business model |
| | 9 | I feel that social media helps me connect directly with consumers and suppliers and better analyze the consumer journey |
| | 10 | I feel that digital technology can maintain beekeeping business market shares in the future and can even create new jobs |
| | 11 | I feel that digital technology helps strengthen the beekeeping business' internal capabilities |
| | 12 | I feel that digital technology helps increase the amount of annual income of the beekeeping business |
| **Business Resilience (Z)** [26,54–56] | | |
| Organizational Robustness | 1 | I can take quick actions to deal with business changes |
| | 2 | I am prepared to manage business challenges that we can foresee |
| | 3 | I can develop new business alternatives to take advantage of the pandemic situation |
| | 4 | I can meet customer needs without disruption |
| | 5 | I continued to maintain the supply network during the pandemic |
| | 6 | I prepared myself when the news of the pandemic broke |
| | 7 | I act in totality in dealing with undesirable business situations |
| Readiness | 8 | I realize and understand that the pandemic has an impact on the beekeeping business |
| | 9 | I took precautions because honey products did not sell well |
| | 10 | I plan and prepare strategies for dealing with future business disruptions |

**Table 1.** *Cont.*

| | | |
|---|---|---|
| **Response** | 11 | I recognized the business threats posed by the COVID-19 pandemic |
| | 12 | I could quickly respond to the negative impact of the pandemic honey sales |
| | 13 | I could provide a solution and recover the beekeeping business from the decreasing consumer awareness after the pandemic |
| | 14 | I have honey products that are considered essential for consumption not only during but also after the COVID-19 pandemic |
| | 15 | I built partnerships with partners during the pandemic and will continue to survive after the pandemic |
| | 16 | I have sufficient internal resources (financial, human resources, production) to deal with unexpected business changes, such as a pandemic |
| | 17 | I can respond appropriately to unexpected disruptions such as pandemics |
| | 18 | I prepare myself as a business manager to deal with a future crisis |
| **Recovery** | 19 | I have good internal and external communication, such as with partners and consumers |
| | 20 | I managed to deal with the crisis caused by the pandemic |
| | 21 | I responded quickly to threats that arose during a pandemic |

### 3.2. Questionnaire Development

This study used a structured questionnaire for collecting data. The questionnaire contains farmers' basic profiles and the main variables' indicators. We constructed the questionnaire from the defined research objectives and scope by reviewing past research. We grouped the constructs based on the research objectives: (1) the effect of DC as an independent variable (sensing, seizing, and reconfiguring) on DT as a dependent variable (IT readiness and strategic alignment); (2) the effect of DT as an independent variable on Business Resilience as a dependent variable (organizational robustness, readiness, response, and recovery); (3) the effect of DC as an independent variable on Business Resilience as a dependent variable; (4) the mediating role of DT as a mediator variable of the link between DC as an independent variable and the Business Resilience as a dependent variable.

Before being distributed to respondents, the questionnaire was verified using validity and reliability tests. Each variable was tested using multivariate Structural Equation Model (SEM) analysis with SmartPLS 3.0 software in the data analyses. The questionnaires were distributed to respondents in hard copy from March to September 2022. The question items are described in Table 1.

Furthermore, making a path model that connects variables and constructs based on theory and logic is the first step in using PLS-SEM [57]. The method's capacity to deal with severe modeling issues, such as atypical data features (for example, nonnormal data) and extremely complicated models, which frequently arise in the social sciences, can be credited for a large portion of the method's expanded utilization. Both observed (indicator) and unobserved (latent) variables are included in structural equation modeling. These variables are divided into measurement models and a structural equation model. Unobserved variables cannot be measured directly; instead, they must be inferred or postulated from the observed variables. Observed variables are those that can be measured. The measurement models describe the indicators used to measure the latent variables [58].

### 3.3. Participants

This study focuses on the case of honey MSMEs in East Java, Indonesia. According to Syamsulbahri [59], micro-enterprises in East Java have adopted technology with a high success rate of around 85%. This indicates to what extent regions with high technology acceptance can apply DT to establish resilient businesses. The research locations with beekeeping business centers were selected based on the information from the East Java Beekeepers Association. They were Kediri, Malang, Probolinggo, and Banyuwangi regencies.

The number of MSME respondents was 388, categorized into five groups based on size. The categorizing of micro, small and medium enterprises refers to the Law of the Republic of Indonesia number 20 of 2008 concerning Micro, Small, and Medium Enterprises. Aside from

that, this study uses asset criteria to identify the business size: micro businesses with assets of IDR <50 million, small businesses with assets of IDR 50 million−500 billion, and medium businesses with assets of IDR 500 million−10 billion. In addition, the categorization also uses the business model criteria, namely family business (FB) and non-family business (NFB). A business is qualified as FB if (1) the family is the major shareholder, (2) the family controls all business activities, and (3) family members as top management [60]. Meanwhile, a business is qualified as NFB if employees are recruited outside the family, even from the surrounding environment [61]. With this, the five sizes and five models in this study are (1) micro-FB, (2) small-FB, (3) small-NFB, (4) medium-FB, and (5) medium-NFB. The multistage random sampling follows these three stages:

1.  Grouping business size and business model
2.  The numbers of samples for each size are different according to empirical conditions
3.  The total sample of 388 beekeeping MSMEs based on the empirical data are as follows: 78 micro-FB respondents, 217 small-FB respondents, 37 small-NFB respondents, and 56 medium-FB respondents.

The SmartPLS test can explain the results of a small sample size of 30–100 [62]. However, it should be noted that the empirical data did not record the existence of medium-NFB and micro-FB respondents. Micro-enterprises in developing countries are often synonymous with FB. Due to limited resources and for efficiency purposes, these businesses employ family members [63].

## 4. Results

Based on an empirical study involving 388 honey enterprises, grouped based on business size and model, the respondents' descriptive profile analysis results are shown in Table 2.

**Table 2.** Respondents' Profiles.

| Description | Micro-FB (78) | | Small-FB (217) | | Small-NFB (37) | | Medium-FB (56) | |
|---|---|---|---|---|---|---|---|---|
| | Number | Percentage | Number | Percentage | Number | Percentage | Number | Percentage |
| **Age** | | | | | | | | |
| 21–30 | 23.0 | 29.5 | 31 | 14.3 | 0 | 0 | 4 | 7.1 |
| 31–40 | 16.0 | 20.5 | 54 | 24.9 | 9 | 24.3 | 14 | 25.0 |
| 41–50 | 31.0 | 39.7 | 94 | 43.3 | 13 | 35.1 | 22 | 39.3 |
| 51–60 | 8.0 | 10.3 | 35 | 16.1 | 13 | 35.1 | 14 | 25.0 |
| 60 above | 0 | 0 | 3 | 1.4 | 2 | 5.4 | 2 | 3.6 |
| **Gender** | | | | | | | | |
| Male | 71.0 | 91 | 210 | 96.8 | 37 | 100.0 | 55 | 98.2 |
| Female | 7.0 | 9 | 7 | 3.2 | 0 | 0 | 1 | 1.8 |
| **Occupation** | | | | | | | | |
| First job | 72.0 | 92.3 | 204 | 94.0 | 37 | 100.0 | 53 | 94.6 |
| Side job | 6.0 | 7.7 | 13 | 6.0 | 0 | 0 | 3 | 5.4 |
| **Capital** | | | | | | | | |
| Self Capital | 7.0 | 9 | 49 | 22.6 | 11 | 29.7 | 14 | 25.0 |
| Formal | 25.0 | 32 | 68 | 31.3 | 16 | 43.2 | 16 | 28.6 |
| Informal | 0 | 0 | 5 | 2.3 | 8 | 21.6 | 0 | 0 |
| Self and formal | 46.0 | 59,0 | 95 | 43.8 | 1 | 2.7 | 26 | 46.4 |
| Self and informal | 0 | 0.0 | 0 | 0 | 0 | 0 | 0 | 0 |
| Formal and informal | 0 | 0.0 | 0 | 0 | 1 | 2.7 | 0 | 0 |
| **Income** | | | | | | | | |
| <10 juta | 31.0 | 39.7 | 70 | 32.3 | 8 | 21.6 | 5 | 8.9 |
| 10–20 juta | 39.0 | 50.0 | 89 | 41.0 | 21 | 56.8 | 13 | 23.2 |
| 20 juta above | 8.0 | 10.3 | 58 | 26.7 | 8 | 21.6 | 38 | 67.9 |

**Table 2.** *Cont.*

| Description | Micro-FB (78) | | Small-FB (217) | | Small-NFB (37) | | Medium-FB (56) | |
|---|---|---|---|---|---|---|---|---|
| | Number | Percentage | Number | Percentage | Number | Percentage | Number | Percentage |
| **Education** | | | | | | | | |
| No education | 0 | 0 | 5 | 6.6 | 0 | 0 | 1 | 1.8 |
| Primary education | 19.0 | 24.4 | 76 | 35.0 | 3 | 8.1 | 13 | 23.2 |
| Junior education | 22.0 | 28.2 | 78 | 35.9 | 5 | 13.5 | 17 | 30.4 |
| Senior education | 35.0 | 44.9 | 51 | 23.5 | 29 | 78.4 | 22 | 39.3 |
| Graduate | 2.0 | 2.6 | 6 | 2.8 | 1 | 2.7 | 3 | 5.4 |
| Postgraduate | 0 | 0 | 1 | 0.5 | 0 | 0 | 0 | 0 |
| **Firm Age** | | | | | | | | |
| <10 years | 15 | 19.2 | 17 | 7.8 | 0 | 0 | 4 | 7.1 |
| 10 years above | 63 | 80.8 | 200 | 92.2 | 37 | 100.0 | 52 | 92.9 |
| **Number of Family** | | | | | | | | |
| <5 | 63 | 80.8 | 162 | 74.7 | 0 | 0 | 40 | 71.4 |
| >5 | 15 | 19.2 | 55 | 25.3 | 37 | 100.0 | 16 | 28.6 |
| **Product demand during crisis** | | | | | | | | |
| Stable | 6 | 8.0 | 15 | 6.9 | 8 | 21.6 | 3 | 5.4 |
| Decrease | 2 | 3.0 | 8 | 3.7 | 7 | 18.9 | 1 | 1.8 |
| Increase | 70 | 89.0 | 194 | 89.4 | 22 | 59.5 | 52 | 92.9 |
| **Business Risk Factor** | | | | | | | | |
| Internal | 7 | 9.0 | 19 | 8.8 | 14 | 37.8 | 14 | 25.0 |
| Eksternal | 68 | 87.2 | 159 | 73.3 | 23 | 62.2 | 33 | 58.9 |
| Internal and External | 3 | 3.8 | 39 | 18.0 | 0 | 0 | 9 | 16.1 |
| **Business Motivation** | | | | | | | | |
| Investation | 25 | 32.1 | 57 | 26.3 | 4 | 10.8 | 16 | 28.6 |
| Business oriented | 36 | 46.2 | 26 | 12.0 | 33 | 89.2 | 28 | 50.0 |
| Both | 17 | 21,8 | 134 | 61,80 | 0 | 0 | 12 | 21.4 |
| **Honey Product Variance** | | | | | | | | |
| Pure honey | 77 | 98.7 | 170 | 78.3 | 32 | 86.5 | 31 | 55.4 |
| Honey product processing | 0 | 0 | 9 | 4.1 | 5 | 13.5 | 10 | 17.9 |
| Both | 1 | 1.3 | 38 | 17.5 | 0 | 0 | 15 | 26.8 |
| **Social Media Used** | | | | | | | | |
| Facebook, instagram | 65 | 83.3 | 207 | 95.4 | 37 | 100.0 | 55 | 98.2 |
| Facebook, isntagram, TikTok | 13 | 16.7 | 10 | 4.6 | 0 | 0 | 1 | 1.8 |
| Facebook, Instagram, TikTok, Youtoube | 0 | 0 | 0 | 0 | 0 | 0 | 0 | 0 |
| **Payment Method** | | | | | | | | |
| Cash | 16 | 20.5 | 64 | 29.5 | 2 | 5.4 | 17 | 30.4 |
| Cashless (mobile banking, e-money) | 18 | 23.1 | 116 | 53.5 | 33 | 89.2 | 21 | 37.5 |
| Both | 44 | 56.4 | 37 | 17.1 | 2 | 5.4 | 18 | 32.1 |
| **Offline Store** | | | | | | | | |
| Yes, available | 62 | 79.5 | 153 | 70.5 | 1 | 2.7 | 21 | 37.5 |
| No | 16 | 20.5 | 64 | 29.5 s | 36 | 97.3 | 35 | 62.5 |

### 4.1. Validity and Reliability Analyses

We carried out several tests to validate the data in the SmartPLS 3.0 SEM analysis. The first is convergent validity, which aims to test the validity of the construct in measuring the independent variables, namely DC and DT, in each typology. The average variant extracted (AVE) value must be above 0.5 [64]. The second is reliability testing through Cronbach's Alpha (CA), with a minimum value limit of 0.5 [65]. Meanwhile, the composite reliability (CR) value from the analysis results shows a value above 0.8, which means that internal reliability is accepted [66]. Third, the loading factor values must be >0.5, applicable to all models [67]. Table 3 presents the results of testing the validity and reliability of the independent variables for each model.

**Table 3.** Validity Test Results.

| Construct. | Second Order | Item | Loading | CA | CR | AVE |
|---|---|---|---|---|---|---|
| | | Model 1 (micro-FB) | | | | |
| | DC | | | 0.758 | 0.857 | 0.667 |
| Sensing | | X1 | 0.759 | | | |
| Seizing | | X2 | 0.874 | | | |
| Reconfiguring | | X3 | 0.813 | | | |
| | DT | | | 0.550 | 0.813 | 0.686 |
| IT-Readiness | | Y1 | 0.877 | | | |
| Strategic Alignment | | Y2 | 0.777 | | | |
| | | Model 2 (small-FB) | | | | |
| | DT | | | 0.837 | 0.901 | 0.753 |
| Sensing | | X1 | 0.828 | | | |
| Seizing | | X2 | 0.867 | | | |
| Reconfiguring | | X3 | 0.906 | | | |
| | DT | | | 0.674 | 0.857 | 0.751 |
| IT-Readiness | | Y1 | 0.906 | | | |
| Strategic Alignment | | Y2 | 0.824 | | | |
| | | Model 3 (small-NFB) | | | | |
| | DC | | | 0.602 | 0.788 | 0.564 |
| Sensing | | X1 | 0.555 | | | |
| Seizing | | X2 | 0.731 | | | |
| Reconfiguring | | X3 | 0.921 | | | |
| | DT | | | 0.904 | 0.954 | 0.912 |
| IT-Readiness | | Y1 | 0.951 | | | |
| Strategic Alignment | | Y2 | 0.959 | | | |
| | | Model 4 (Medium-FB) | | | | |
| | DC | | | 0.843 | 0.903 | 0.758 |
| Sensing | | X1 | 0.871 | | | |
| Seizing | | X2 | 0.924 | | | |
| Reconfiguring | | X3 | 0.813 | | | |
| | DT | | | 0.644 | 0.848 | 0.737 |
| IT-Readiness | | Y1 | 0.878 | | | |
| Strategic Alignment | | Y2 | 0.839 | | | |

*4.2. Structural Measurement Model*

The results of data analysis using SmartPLS 3.0 show that the SRMR values in Model 1, Model 2, Model 3, and Model 4 are 0.094, 0.089, 0.083, and 0.097, respectively. The values of SRMR less than 0.10 are generally considered favorable [68]. Meanwhile, the NFI values were 0.830, 0.768, 0.794, and 0.717, respectively. The NFI value must be between 0 and 1; the closer the value to 1, the better the model is [69]. This study shows that in Model 1 (micro-FB), DT can be explained by DC at 42.9%, whereas DT does not affect resilience. In Model 2 (small-FB), DT can be explained by DC at 59.8%, and Business Resilience can be explained by DT at 69.1%. In Model 3 (small-NFB), DT can be explained by DC at 64.3%, whereas the impact of DT on resilience was not observed. Lastly, in Model 4 (medium-FB), DT can be explained by DC at 36.2%, and Business Resilience can be explained by DT at 70.3%.

Figure 2 and Table 4 is a structural model showing the effect of the independent variables on the dependent variable. The results of SEM analysis with smartPLS 3.0 show differences in the influence of business size and business model. Model 1 shows that Hypothesis 1 (H1) is accepted (*t*-value = 11.692), suggesting the effect of DC has an effect on DT in Micro-FB. Meanwhile, Hypothesis 2 (H2) is rejected (*t*-value = 0.110), suggesting no effect of DT on resilience in Micro-FB. Meanwhile, Hypothesis 3 (H3) is accepted (*t*-value = 2.442), suggesting that DC affects resilience. In this model, DT did not show a function as a mediator variable between DC and resilience, so Hypothesis 4 (H4) was rejected (*t*-value = 0.111). Furthermore, we also estimate the model based on firm catogeries and it was presented in Figure 2. The

Micro-FB Empirical Model, Figure 3. For Small-FB Empirical Model, Figure 4. For Small-NFB Empirical Model, and Figure 5. for Medium-FB Empirical Model.

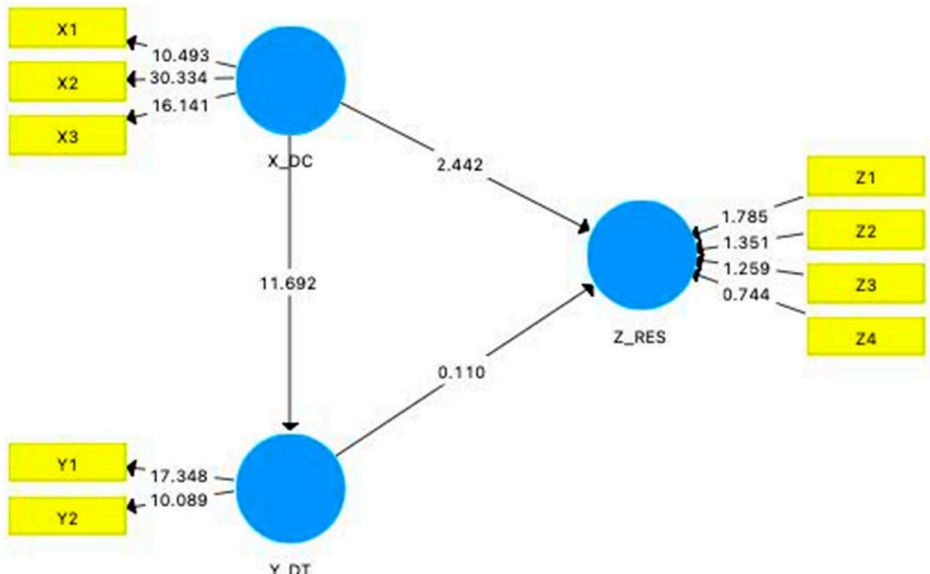

**Figure 2.** The Micro-FB Empirical Model.

**Table 4.** Hypothesis Testing.

| Hypothesis | Relationship | Std.Beta | O/STDEV | *p*-Values | Decision |
|---|---|---|---|---|---|
| **Model 1 (Micro-Family Business)** | | | | | |
| Direct effect | | | | | |
| H1 | DC > > > DT | 0.655 | 11.692 | 0.000 | Supported |
| H2 | DT > > > RES | 0.024 | 0.110 | 0.912 | Rejected |
| H3 | DC > > > RES | 0.522 | 2.442 | 0.015 | Supported |
| Indirect effect | | | | | |
| H4 | DC > DT > RES | 0.016 | 0.111 | 0.912 | Rejected |
| **Model 2 (Small-Family business)** | | | | | |
| Direct effect | | | | | |
| H1 | DC > > > DT | 0.773 | 34.512 | 0.000 | Supported |
| H2 | DT > > > RES | 0.627 | 7.865 | 0.000 | Supported |
| H3 | DC > > > RES | 0.245 | 3.289 | 0.001 | Supported |
| Indirect effect | | | | | |
| H4 | DC > DT > RES | 0.189 | 3.270 | 0.001 | Supported |
| **Model 3 (Small-Non Family Business)** | | | | | |
| Direct effect | | | | | |
| H1 | DC > > > DT | 0.802 | 2.603 | 0.010 | Supported |
| H2 | DT > > > RES | 0.033 | 0.114 | 0.909 | Rejected |
| H3 | DC > > > RES | 0.816 | 1.231 | 0.219 | Rejected |
| Indirect effect | | | | | |
| H4 | DC > DT > RES | 0.654 | 1.476 | 0.141 | Rejected |
| **Model 4 (Medium-Family Business)** | | | | | |
| Direct effect | | | | | |
| H1 | DC > > > DT | 0.601 | 10.080 | 0.000 | Supported |
| H2 | DT > > > RES | 0.455 | 4.613 | 0.000 | Supported |
| H3 | DC > > > RES | 0.482 | 4.868 | 0.000 | Supported |
| Indirect effect | | | | | |
| H4 | DC > DT > RES | 0.274 | 3.895 | 0.000 | Supported |

Note: dynamic capability (DC), digital transformation (DT), Business Resilience.

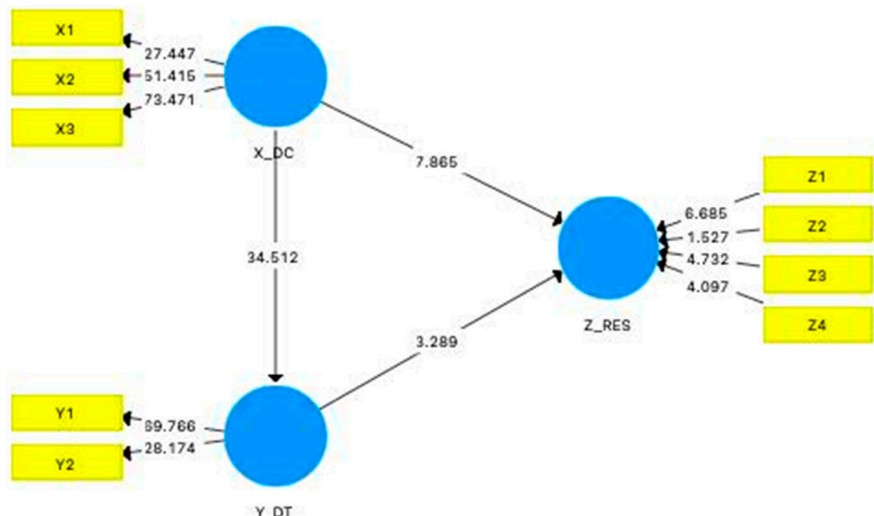

**Figure 3.** The Small-FB Empirical Model.

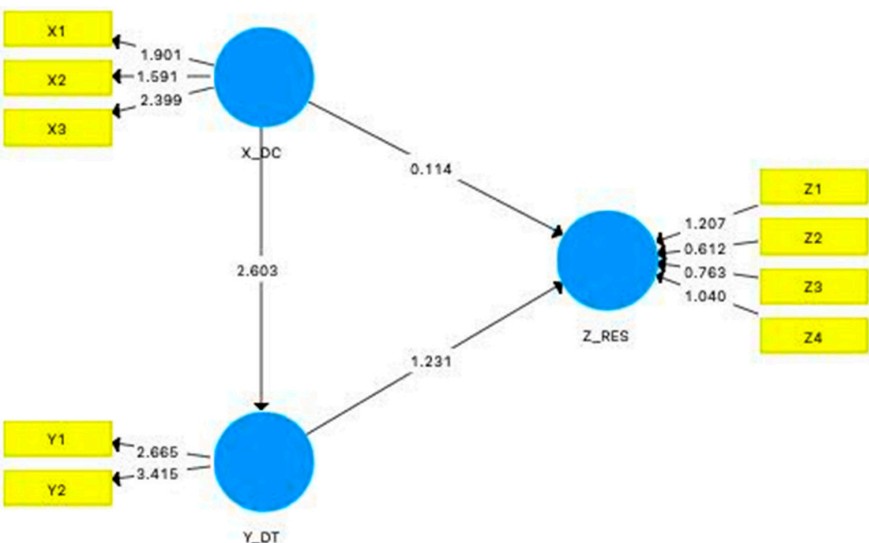

**Figure 4.** The Small-NFB Empirical Model.

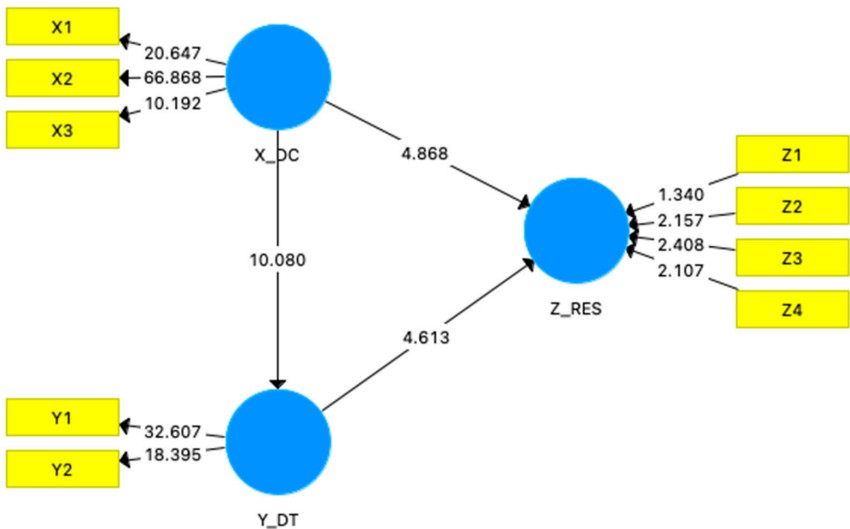

**Figure 5.** The Medium-FB Empirical Model.

Model 2 shows that DC affects DT in Small-FB, so Hypothesis 1 (H1) is accepted (t-value = 34.512). DT is also proven to affect Business Resilience, so Hypothesis 2 (H1) is accepted (t-value = 7.865). Likewise, there is a direct effect of DC on Business Resilience, so Hypothesis 3 (H3) is accepted (t-value = 3.289). Unlike in Model 1, DT is proven to show a mediating role between DC and resilience in this model, which means that Hypothesis 4 (H4) is accepted (t-value = 3.270).

As for small-NFB cases, Model 3 shows that DC affects DT, so Hypothesis 1 (H1) is accepted (t-value = 2.603). However, DT does not affect resilience, so Hypothesis 2 (H2) is rejected (t-value = 0.114). Likewise, DC also does not affect resilience, so Hypothesis 3 (H3) is rejected (t-value = 1.231). Similar to Model 1, DT does not mediate DC and resilience, so Hypothesis 4 (H4) is rejected (t-value = 1.476).

Model 4 for medium-FB shows that DC directly affects DT, so Hypothesis 1 is accepted (H1) (t-value = 10.080). In addition, DT also affects resilience, so hypothesis 2 (H2) (t-value = 4.613) is accepted. This model also shows that DC positively affects resilience, so Hypothesis 3 (H3) is accepted (t-value = 4.868). In this model, DT shows a mediating role between DC and resilience, so Hypothesis 4 (H4) is accepted (t-value = 3.895).

## 5. Discussion

This study examines the relationship between DC and resilience and the mediating role of DT in different business sizes and models, i.e., micro-FB, small-FB, small-NFB, and medium-FB.

### 5.1. Micro-FB

In Micro-FB, DC had a positive effect on DT. With limited resources in micro-enterprises, businesses need to manage resources efficiently through DC. To do this, they need sensitivity to market changes, the ability to respond quickly to opportunities, and the ability to reconfigure limited internal resources and outsource business factors they lack. With this, micro-enterprises can achieve DT. However, the model shows that DT does not determine business resilience. A possible explanation is that DT requires highly skilled human resources and innovation not only in the digital field but also in management. Meanwhile, the human resources of Micro-FB observed in this study were dominated by family members aged between 41 and 50 years who may not be agile in adopting technology. In addition, the internal structure does not support the exploration of changes in the business model to suit customer needs. Hence, business innovation is not only derived from technological capabilities but also innovation within the organization.

This finding shows that the limitation in micro businesses contrasts with the condition of large companies where resources are abundant. For example, they can recruit technology-adaptive human resources. Vathanophas et al. [70] argued that when companies with limited resources try to innovate their technology and change their business models, the effort may result in inefficient use of resources. The choices are to adopt technology quickly or to adopt technology effectively to reach an optimum result and positively impact performance [71].

This finding also shows the weakness of the FB model in the beekeeping business. A family culture in business management tends to be traditional, which influences the decision to change the business model. This is in accordance with the results of a knowledge study by Cabrera-Suárez et al. [72], arguing that business successors accumulate the knowledge of previous business holders as a form of cultural and knowledge transfer. Meanwhile, external influences are needed in DT decision-making, but Micro-FB's external influences may only be limited to friends and family members [73]. In other words, Micro-FB does not perform optimally in DT processes due to limited resources and knowledge. The business culture, problems of family involvement, and problems among generations of business owners may lead to inflexibility and resistance to changes in entrepreneurial leadership [74]. Therefore, the role of DT does not mediate DC to achieve a resilient business.

### 5.2. Small-FB

In Small-FB, DC positively affects DT, suggesting that DC is a critical success factor for DT. Sensing, seizing, and reconfiguring can support Small-FB in changing its business model according to market changes. As such, small-FB can continue to build relationships with external parties to compensate for their limited resources. This is in line with the study by Matarazzo et al. [75], maintaining that DC begins with sensing, a starting point for entrepreneurs to initiate DT, where products and services are adjusted according to customer needs. In other words, DC encourages value co-creation in companies. DC then results in the ability to reconfigure internal and external resources, which is highly crucial for small businesses with limited capital [76].

On the other hand, when a company in the Small-FB category decides to perform DT, it will probably achieve resilience. Even though the age group between micro-FB and small-FB tends to be the same, the internal resources of small-FB are larger than those of micro-FB. Indeed, the respondents' profiles observed in this study show that the capital of those in small-FB (self-capital) is higher than those in micro-FB. In addition, small-FB consumers are more aware of technological change, as shown in a large number of digital payments or cashless transactions. This is in line with a study by Kellermanns et al. [77], showing that in FB innovation, other family members are cooperative enough to provide support. This means that the top management can optimize the innovation to increase profits. Yet, this finding also supports the absolute advantage theory by having better DC and DT by maximizing their profit [40]. Therefore, DT's decisions in Small-FB play an essential role in mediating DC in achieving business resilience. This case indicates that FB's strength lies in its close ties and sense of business ownership. Likewise, a study by Sharma [78] shows that FB is proven to be effective in increasing internal resources. Miller et al. [79] show that the futuristic nature of FB (long-term commitment to the business) makes it easier for them to improve business performance.

### 5.3. Small-NFB

In the Small-NFB cases, DC has a significant effect on DT. The entrepreneurs were able to deal with changes as they could sense dynamics, recognize opportunities, and reconfigure external and internal resources to adopt DT. However, even though they have performed DT, resilience is hard to achieve. Considering the age range in Small-NFB, which is 41 years and over, DT may not be optimal, as in Micro-FB. Moreover, the sense of belonging to the NFB is lower because the NFB emphasizes professionalism in achieving business goals.

Kellermanns et al. [77] show that family reciprocal actions and attitudes could increase a sense of belonging and create common goals among individuals, which is not the case in NFB, as individual interest is more dominant. Thus, DC does not foster resilience among Small-NFB companies. The intangible assets in DC and DT have not been able to optimize the NFB business, so DT cannot act as a mediator between DC and resilience. This is in line with Amann et al. [80] study, showing that FBs were more resilient during the economic crisis and afterward than NFBs, and recovered faster, with better business performance and a more robust financial structure. This is because family investments tend to be bigger and ties are stronger, so they can mobilize resources better than NFBs. This finding is also in line with a study by Joosse et al. [81], showing that the main obstacle for NFB is the high initial capital. There may be deficiencies in the reconfiguration of internal and external resources. In addition, NFBs tend to be careful in making decisions to avoid risks, so they may experience more resource shortages than FBs [82]. Since DC and DT did not essentially contribute to small NFBs' resilience, other factors could be considered to maintain their resilience, such as entrepreneur orientation [83], human and social capital [84], and market orientation [6].

### 5.4. Medium-FB

In the case of Medium-FB, DC proved to have a positive effect on DT and resilience. A possible explanation is that the resources are more extensive, making it easier for en-

trepreneurs to take strategic actions to explore new opportunities and resources [85]. Indeed, resources are essential to encourage FBs to modernize their organizations. Therefore, DT is crucial in mediating DC and resilience, suggesting that entrepreneurs must perform DT to achieve a resilient business.

This finding is in line with a study by Duran et al. [86], which shows that FBs tend to be more innovative than NFBs because of their long-term investment and commitment to the business's future [87]. Referring to the respondents' profiles, the ability to build relationships in the Medium-FB group tends to be better than the businesses of other sizes and models. In addition, they can create opportunities and are more agile; for example, by selling honey products online to reach more customers rather than spending resources to open an offline store. In addition, this group has higher education levels (bachelor's degree) than the other business sizes, which tend to be more adsorptive and adaptive than NFB and Micro-FB.

In sum, the findings have shown that the role of DC is vital for beekeeping MSMEs in Indonesia by encouraging them to perform DT. However, not all MSMEs can achieve a resilient business with the achievement of DC and DT. Based on the business models, the insights into how FB and NFB are affected by DC and DT can provide the groundwork for future researchers to understand more about the ability of MSMEs to shift models and adapt to market changes.

*5.5. Theoretical Contributions*

This study contributes to the existing literature five-fold. First, this study adds novelty to the existing theory related to DT (IT-readiness and strategic alignment) in the beekeeping business case of different sizes and models, i.e., micro-FB, small-FB, small-NFB, and medium FB, indicating different results. Second, the findings have shown that DT plays an important role in mediating DC (sensing, seizing, and reconfiguring) with resilience in the case of medium-FB. Third, this study has illustrated how DT can encourage small-FB and medium-FB to achieve resilience. Fourth, this study has proven that DC can foster resilience in the case of micro-FB and small-FB. Finally, the finding also informs that business sizes and models are important considerations to encourage MSMEs to implement DT in order to achieve resilience.

*5.6. Practical Implications*

On a practical level, this study has shown that DC can impact resilience, mediated by DT. First, DC (sensing, seizing, and reconfiguring) helps all business sizes and models to achieve DT across different management conditions. Second, human resource capacity determines the ability to recognize changes, seek opportunities to meet customer needs, and reconfigure internal and external resources to make DT decisions. Previous studies Deng et al. and Grimpe et al. [88,89] have shown that adsorptive and adaptive capabilities increase technology acceptance and make organizational management more agile for transformation [90]. Second, companies must have IT readiness and strategic alignments to achieve resilience. For example, Small-FB and Medium-FB have a futuristic orientation, which benefits their business, but this orientation needs to be coupled with technology adoption and business strategy [91,92].

Therefore, top management across sizes and models should consider starting with their intangible assets to increase their business resilience because each business typology has different business characteristics. This can be done by improving firm technology innovation which is still employed the traditional way. Also, increase and maintain customer trust, and produce honey products based on the customer segment which is overlooked in Indonesia. The government can improve firm digital transformations by providing business-related digitalization such as marketing applications, e-money, and information that focuses on specific firm categories.

*5.7. Limitation*

This study is limited to the case of honey SMEs in East Java, Indonesia, so it cannot be generalized to explain intangible assets in other food and non-food cases and business typologies other than size and model. In addition, the different number of samples in each group has only allowed the structural model to cover four typologies of the beekeeping businesses in East Java. This study is also limited to MSMEs and the FB and NFB models. The minimum firm age we use as a sample is also limited to 10 years. Referring to the results of previous studies, education and technology adoption could act as mediating variables that can influence the relationship between the DC and DT, as well as between DT and resilience.

Nonetheless, this study has provided empirical evidence on the mediating role of DT in the impact of DC on resilience, which is apparent in Medium-FB. This suggests an opportunity to explore other livestock products, which tend to be dynamic in the face of a crisis. With the impact of the pandemic that has been massive in the past few years, researchers have explored the relationships between DC, DT, and resilience to deal with the shocks and the accelerating technological change in society. Thus, this study can be tested in other strategic commodities that contributed to the recovery in the MSME sector and the economy in general. In fact, this study can also be applied to other uncertainties, such as inflation, recession and climate change [93]. Future research will benefit from replicating the model in the food and non-food sectors in developing or developed countries.

## 6. Conclusions

The results of this study can be generalized to the national level to inform decision-making regarding the intangible assets of MSME livestock products in developing countries. The findings are also relevant to other livestock products, which tend to be dynamic during a crisis.

The relationships between DC and DT mediated by DC are not only theoretical. On a practical level, they can be implemented in small, medium, and large companies to deal with future uncertainties. DC is proven essential in encouraging DT, as shown empirically in the tests on four SMSE typologies in Indonesia. Aside from that, achieving a resilient business requires sound decision-making and effective implementation of DT, as exemplified in Small-FB and Medium-FB in this study. This is because DT does not only cover capabilities in the technology field but also flexibility and agility that allows strategic management to take place. On the other hand, DT plays a significant role in connecting DC and resilience, as observed in Medium-FB. As such, achieving a resilient business requires technological readiness and strategic alignment.

**Author Contributions:** Conceptualizations, methodology and validation, J.A.P., H.T. and M.S.R.; investigation, resources, data curation J.A.P., H.T., M.S.R., H.N.U. and D.M., formal analysis, writing—original draft preparation, J.A.P., writing—review and editing, H.T., B.H., M.S.R. and D.M.; supervision B.H. and H.N.U. All authors have read and agreed to the published version of the manuscript.

**Funding:** Doctoral Dissertation Research Grant, Ministry of Education and Culture of the Republic of Indonesia. No. 1071.11/UN10.C10/TU/2022.

**Institutional Review Board Statement:** Not applicable.

**Informed Consent Statement:** Not applicable.

**Data Availability Statement:** The data presented in this study are available on request from the corresponding author. The data are not publicly available due to ethical reasons.

**Acknowledgments:** We would like to thank the Doctoral Dissertation Research Grant Office at the Ministry of Education and Culture of the Republic of Indonesia for funding and supporting this research.

**Conflicts of Interest:** The authors declare no conflict of interest.

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
