# Peer review of "Do Dynamic Capabilities and Digital Transformation Improve Business Resilience during the COVID-19 Pandemic? Insights from Beekeeping MSMEs in Indonesia"

_sustainability, doi:10.3390/su15031760_

Round 1
Reviewer 1 Report
The representation of the Abstract is apparent.
Perfect explanation of the Introduction section with all the background concepts discussed thoroughly. Could have added more details about honey enterprises (Beekeeping MSMEs) in Indonesia and the rest of the world.
Explicit discussion of theoretical background with hypothesis development. The hypothesis is clearly explained with the help of the research framework.
Thorough preparation of questionnaire with variables and indicators.
Subheadings 3.2 and 3.3 are repeated on page number 7. Check these contents and accordingly change subsequent subheadings.
Results and discussion are explained in detail with reliability analysis and structural measurement model for all the models.
The last citation under References needs to number suitably.
Overall contents discussed in the paper are clear and easy to understand, with good supporting results and discussion.
Author Response
Dear Reviewer:
Thank you for giving us the opportunity to submit a revised draft of our manuscript. We appreciate the time and effort that you and the reviewers have dedicated to providing your valuable feedback on our manuscript. We are grateful to the reviewers for their insightful comments on our paper. We have been able to incorporate changes to reflect most of the suggestions provided by the editorial team. We have highlighted the changes within the manuscript.
Reviewer 1
Point 1: The representation of the Abstract is apparent.
Response: Many thanks for your kind response
Point 2: Perfect explanation of the Introduction section with all the background concepts discussed thoroughly. Could have added more details about honey enterprises (Beekeeping MSMEs) in Indonesia and the rest of the world.
Response: Thanks so much, we have address this issue in line 101
Point 3: Explicit discussion of theoretical background with hypothesis development. The hypothesis is clearly explained with the help of the research framework.
Response: Many thanks for your kind response
Point 4: Thorough preparation of questionnaire with variables and indicators.
Response: Thank you for your suggestion, we added the explanation in line 246, and it also related with table 1.
Point 5: Subheadings 3.2 and 3.3 are repeated on page number 7. Check these contents and accordingly change subsequent subheadings.
Response: The subsection 3.3 have been revised
Point 6: Results and discussion are explained in detail with reliability analysis and structural measurement model for all the models.
Response: Many thanks for your kind response
Point 7: The last citation under References needs to number suitably.
Point 8: Overall contents discussed in the paper are clear and easy to understand, with good supporting results and discussion.
Response: Many thanks for your kind response
Reviewer 2 Report
My suggestions to improve this paper is in attachment

Author Response
Dear Reviewer:
Thank you for giving us the opportunity to submit a revised draft of our manuscript. We appreciate the time and effort that you and the reviewers have dedicated to providing your valuable feedback on our manuscript. We are grateful to the reviewers for their insightful comments on our paper. We have been able to incorporate changes to reflect most of the suggestions provided by the editorial team. We have highlighted the changes within the manuscript.
Reviewer 2.
- Regarding for improving the Theoretical Framework and supporting views, it is better to add below papers to mentioned articles:
- Gholizadeh, salar. Mohammmadkazemi, Reza. (2022). International Entrepreneurial Opportunity: A systematic review, meta-synthesis, and future research agenda, Journal of International Entrepreneurship, Vol 20, Issue 1 (March 2022)
- Mohammmadkazemi, Reza. Nikraftar, H. Yadollahi Farsi, J. Ahmadpour, M. (2019). The Concept of International Entrepreneurial Orientation in Competitive Firms: A Review & A Research Agenda. International Journal of Entrepreneurship, Volume 23, Issue 3, 2019.
- Rogier van de Wetering, (2022), The role of enterprise architecture-driven dynamic capabilities and operational digital ambidexterity in driving business value under the COVID19 shock, Heliyon, Volume 8, Issue 11, 2022,
- Suja Chaulagain, Melissa Farboudi Jahromi, Xiaoxiao Fu, (2021). Americans' intention to visit Cuba as a medical tourism destination: A destination and country image perspective, Tourism Management Perspectives, Volume 40, 100900,
Response : Many thanks for the usefull suggestion, it can improve the value of our manuscript, we have add the literature mentioned above this issue on 158 and 173
After adding above suggested references, It is recommended to author/authors to rewrite the part of” “Discussion and conclusion”. I appreciate the kindness of the Editor in helping to improve the manuscripts. Please do not hesitate to contact me if there are any questions. Sincerely Yours,
Response : Thank you so much, the discussion was revised on line 432
Reviewer 3 Report
Dear Authors,
despite this is an interesting study, the current version has to be improved. In particular, the role of resilience is not deepened as much as it needs. From this point of view few references are dedicated to resilience and also discussion must be more connected to it. Specify better, moreover: the gaps you are covering, the methodology you are applying (give also some references about SEM). Practical contribution, furthermore, must be deepened.
Good luck!
Author Response
Dear Reviewer:
Thank you for giving us the opportunity to submit a revised draft of our manuscript. We appreciate the time and effort that you and the reviewers have dedicated to providing your valuable feedback on our manuscript. We are grateful to the reviewers for their insightful comments on our paper. We have been able to incorporate changes to reflect most of the suggestions provided by the editorial team. We have highlighted the changes within the manuscript.
Reviewer 3
Dear Authors,
despite this is an interesting study, the current version has to be improved. In particular, the role of resilience is not deepened as much as it needs. From this point of view few references are dedicated to resilience and also discussion must be more connected to it. Specify better,
Response : Many thanks for the useful suggestion, it can improve the value of our manuscript, we have address this issue on 287, 453
moreover: the gaps you are covering, the methodology you are applying (give also some references about SEM). Practical contribution, furthermore, must be deepened.
Response : Thanks for your useful suggestion, we add the explanation on line 91-96, and 287,
Good luck!
Round 2
Reviewer 3 Report
Dear Authors,
many thanks for the renewed version of the paper. I find that the concept of business resilience is not deepened yet, you should focus on it in the introduction. Moreover I still find weak the paragraph on practical implications, deepen it.
Author Response
Many thanks for your suggestion,
- many thanks for the renewed version of the paper. I find that the concept of business resilience is not deepened yet, you should focus on it in the introduction. Moreover I still find weak the paragraph on practical implications, deepen it.
Responses
We added the resilience dissucsion in line 37, and line 270 for the practical implication